

# Inverse stochastic-dynamic models for high-resolution Greenland ice-core records

Niklas Boers[1], Mickael D. Chekroun[2], Honghu Liu[3], Dmitri Kondrashov[2,4], Denis-Didier Rousseau[1,5], Anders Svensson[6], Matthias Bigler[7], and Michael Ghil[1,2]

[1]Geosciences Department and Laboratoire de Météorologie Dynamique (CNRS and IPSL), École Normale Supérieure and PSL Research University, Paris, France.
[2]Department of Atmospheric and Oceanic Sciences and Institute of Geophysics and Planetary Physics, University of California, Los Angeles, USA.
[3]Department of Mathematics, Virginia Polytechnic Institute and State University, Blacksburg, USA.
[4]Institute of Applied Physics of the Russian Academy of Sciences, Nizhny Novgorod, Russia.
[5]Lamont-Doherty Earth Observatory, Columbia University, Palisades, New York, USA.
[6]Centre for Ice and Climate, University of Copenhagen, Copenhagen, Denmark.
[7]Physics Institute and Oeschger Center for Climate Change Research, University of Bern, Bern, Switzerland.

*Correspondence to:* Niklas Boers (nboers@lmd.ens.fr)

**Abstract.** Proxy records from Greenland ice cores have been studied for several decades, yet many open questions remain regarding the climate variability encoded therein. Here, we use a Bayesian framework for inferring inverse, stochastic-dynamic models from $\delta^{18}O$ and dust records of unprecedented, subdecadal temporal resolution. The records stem from the North Greenland Ice Core Project (NGRIP) and we focus on the time interval 59 ka–22 ka b2k. Our model reproduces the dynamical characteristics of both the $\delta^{18}O$ and dust proxy records, including the millennial-scale Dansgaard-Oeschger variability, as well as statistical properties such as probability density functions, waiting times and power spectra, with no need for any external forcing. The crucial ingredients for capturing these properties are (i) high-resolution training data; (ii) cubic drift terms; (iii) nonlinear coupling terms between the $\delta^{18}O$ and dust time series; and (iv) non-Markovian contributions that represent short-term memory effects.

## 1 Introduction

Data-driven stochastic difference equation models have recently been successfully applied to a wide range of climatic phenomena (Kondrashov et al., 2005, 2006; Kravtsov et al., 2005, 2009). The striking success of these authors' Empirical Model Reduction (EMR) approach in reproducing dynamical and statistical properties of the underlying dynamical systems has been explained by embedding EMR models in the larger class of multilayer stochastic models (MSMs); the latter models, in turn, were shown to be solidly grounded in the Mori-Zwanzig (MZ, (Zwanzig, 1964; Mori, 1965)) formalism of statistical physics (Kondrashov et al., 2015). In addition to enhancing our understanding of the geophysical systems under consation, EMR-MSM models have also been shown to be well suited for predictive purposes; e.g., they have considerable skill in predicting





certain key variables associated with the El Niño–Southern Oscillation (Barnston et al., 2012; Chekroun et al., 2011a) and the Madden-Julian Oscillation (Kondrashov et al., 2013).

In general terms, stochastic-dynamic models are derived by approximating the discrete-time divided differences of observed time series by a deterministic function $\mathbf{F}$ plus residual noise (e.g. Hasselmann, 1976):

$$5 \quad \frac{\Delta \mathbf{x}_i}{\Delta t_i} \approx \mathbf{F}(\mathbf{x}_i) + \eta_i \; ; \tag{1}$$

here $\mathbf{x}_i$ are empirical observations, $\Delta \mathbf{x}_i = \mathbf{x}_{i+1} - \mathbf{x}_i$, and $\Delta t_i = t_{i+1} - t_i$ denote the time spans from one observation to the next. The specific functional form of $\mathbf{F}$ may use some a priori knowledge of the system under study, while the parameters of the proposed model are always inferred by training it on a given set of time series produced by the system. In this sense, the approach is semi-empirical, rather than being entirely hypothesis-free. Most existing methodologies for empirical-model derivation are based on least-squares fitting to determine optimal parameters for $F$.

In the present study, we are specifically interested in fitting low-dimensional stochastic-dynamic models to high-resolution time series of two paleoclimatic proxy records, namely the $\delta^{18}$O ratios and dust concentrations obtained by the North Greenland Ice Core Project (NGRIP; Andersen et al., 2004); see Fig. 1. Low-dimensional conceptual models have a long history in paleoclimate (e.g. Källén et al., 1979; Le Treut and Ghil, 1983; Saltzman and Maasch, 1990; Ghil, 1994; Tziperman et al., 2006; De Saedeleer et al., 2013). Such models typically incorporate a few global or regional climate variables, such as global temperature and ice-sheet volume, nonlinear interactions among these, and astronomical forcing subject to possible stochastic fluctuations (Saltzman and Maasch, 1991; Crucifix and Rougier, 2009).

In contrast, we choose here a data-driven, stochastic-dynamic approach: We intend to find a system of stochastic differential equations (SDEs) to simulate time series that reproduce the statistical and dynamical properties of the observed $\delta^{18}$O and dust time series, and do so without taking into account exogenous astronomical forcing. The main issue with the naive approach of equation (1) is that the noise term $\eta$, which represents the unobserved variables, will typically be correlated with the state vector $\mathbf{x}$. To overcome this problem, we adapt the recently developed non-Markovian data-driven closure models introduced by Kondrashov et al. (2015) as follows.

Given a multivariate, low-dimensional time series $\mathbf{x}(t)$ of partial observations of a much higher-dimensional system, the MZ formalism yields the abstract Generalized Langevin Equation (GLE)

$$\frac{\mathrm{d}\mathbf{x}}{\mathrm{d}t} = \mathcal{F}(\mathbf{x}) + \int_0^t \mathcal{G}(t, s; \mathbf{x}(s))\mathrm{d}s + \eta(t) \,. \tag{2}$$

In our application, $\mathbf{x}$ is the two-vector of $\delta^{18}$O and $\log(\mathrm{dust})$, while the much larger system is the climate system. The first term $\mathcal{F}(\mathbf{x})$ is Markovian and it accounts for the nonlinear self-interactions among the observed variables, while the non-Markovian integral term accounts for the cross-interactions between the observed and unobserved variables; in this formulation, the latter are not present in $\mathcal{F}(\mathbf{x})$. This non-Markovian integral term involves the past of the observed variables and thus introduces memory effects into the closed system. The term $\eta(t)$ accounts for the stochastic forcing that is now uncorrelated with $\mathbf{x}$, as guaranteed by the MZ formalism. However, the noise term is not necessarily white in time or space; see (Kondrashov et al., 2015) for a detailed derivation of the GLE approach, and Appendix A for a sketch of the main ideas and more technical details.





The temporal evolution of both the $\delta^{18}$O and dust time series indicates that there exist two alternative, relatively steady states for the underlying dynamical system, namely the colder stadials and the warmer interstadials (Dansgaard et al., 1993; Rasmussen et al., 2014). Transitions from the stadials to the interstadials occur very abruptly, within several decades, during the so-called Dansgaard-Oeschger events (Johnsen et al., 1992; Dansgaard et al., 1993; Ditlevsen et al., 2005), while transitions in

the opposite direction are characterized by a comparably slow relaxation process that may last centuries to millennia, depending on the specific event.

This bistability suggests using a system of two coupled SDEs with a double-well potential as a model of the processes generating the two time series of $\delta^{18}$O ratios and dust (Ditlevsen, 1999; Ditlevsen and Ditlevsen, 2009). In addition, we will take into account here possible memory effects in the climate system (Bhattacharya et al., 1982; Ghil et al., 2015) by including

explicit non-Markovian terms in the model. In particular, these memory terms are included in order to reproduce the sharp transitions from the stadials to the interstadials, and rather smooth transitions from the interstadials back to the stadials. For these data, we thus propose a two-dimensional stochastic delay differential equation as approximation of the GLE (2):

$$\mathrm{d}\mathbf{x} = \Big\{ A + \sum_{s=0}^{d} B_s \mathbf{x}(t - s\tau) + C(\mathbf{x},\mathbf{x}) + D(\mathbf{x},\mathbf{x},\mathbf{x}) \Big\} \mathrm{d}t + Q\,\mathrm{d}W(t). \tag{3}$$

Here $\mathbf{x}$ is the two-dimensional time series of $\delta^{18}$O and dust, which is sampled at time steps $t_i$ in the NGRIP ice core. The

model has a cubic drift term and retarded, non-Markovian arguments in the linear terms. The matrix $Q$ denotes the Cholesky decomposition of the covariance matrix of the noise, and $W(t)$ denotes a multidimensional Wiener process. Model parameters will be inferred using Maximum Likelihood Estimation (MLE) and, for comparison, ordinary least-squares fitting. It will turn out that both approaches are in fact equivalent for the specific optimization problem proposed here; see Sec. 2.2 for further details and the explicit version of this SDE that we use in practice.

Time series simulated by our empirical model will be compared to the original time series in terms of statistical properties, such as the probability density functions (PDFs) of the time series, their power spectra, and the average waiting time between sharp transitions from stadials to interstadials. Furthermore, we will test the relevance of the different model ingredients, such as the nonlinear terms, the memory and the coupling terms, using Bayesian model selection criteria.

The general potential of Bayesian parameter inference such as MLE has recently been discussed for stochastic-dynamic

climate models from incomplete data (Peavoy et al., 2015). A Bayesian framework to compare different types of models has also recently been used for the specific case of the NGRIP $\delta^{18}$O record, including a double-well potential model, a relaxation oscillator, and two versions of a mixture of locally linear stochastic models. Based on the Bayesian information criterion, it was concluded there that the relaxation oscillator model is best supported by the observations (Kwasniok, 2013). Furthermore, a comprehensive Bayesian approach was employed to infer parameters for a double-well potential model from the NGRIP $\delta^{18}$O

record (Krumscheid et al., 2015). Most recently, a double-well potential model was compared to a relaxation oscillator model with different external forcings using comparative Bayesian statistics (Mitsui and Crucifix, 2016). There, the conclusion was that the oscillator model is the more likely model candidate given the data, and in particular that external forcing in terms of the ice volume significantly improves the statistical model. The latter three studies, however, used a lower-resolution version





of the record, and did not consider either memory terms or coupling between the $\delta^{18}$O record and the dust record, as will be done here.

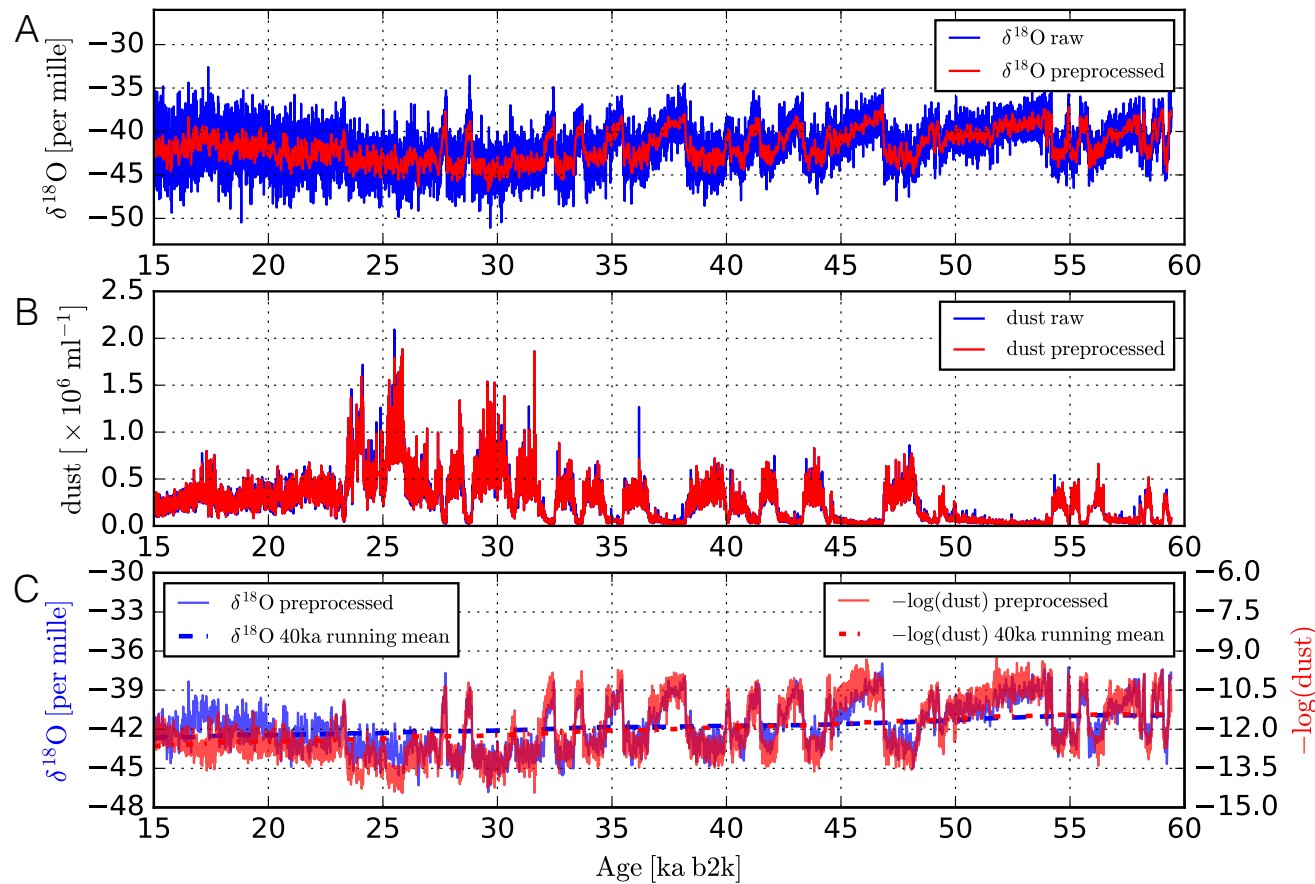

**Figure 1. Time series of $\delta^{18}$O and dust from the NGRIP record for the time interval 15 ka–59 ka b2k; following paleorecord use, the time axis points from the present (at left) towards the past (at right).** (A) Raw (blue) and preprocessed (red) $\delta^{18}$O time series. (B) Raw (blue) and preprocessed (red) dust time series. The two records are visually indistinguishable; the blue is visible only where it exceeds the red. (C) SSA-smoothed $\delta^{18}$O (blue) and $-\log(\text{dust})$ (red), together with their 40-ka running means (dashed lines). Note the strong covariability between the two preprocessed time series during the interval 59 ka–22 ka b2k; the Pearson's correlation coefficient on this interval equals 0.86. For details on the preprocessing of the records see Sec. 2.1

.





## 2  Data and Methods

### 2.1  High-resolution NGRIP data

We employ proxy records of $\delta^{18}$O ratios and dust concentrations from the same core at the NGRIP drilling site. The $\delta^{18}$O ratios were regularly sampled every 5 cm, while the dust concentrations were sampled at a resolution of 1 mm (Ruth et al., 2002,

2003). The dust record was resampled here to the lower 5-cm resolution for consistency with the $\delta^{18}$O record. The proxy time series for the two variables have a common chronology, referred to as GICC05 (Svensson et al., 2008), for the time interval starting at approximately 59 ka b2k. Here, 1 ka = 1000 a and "b2k" refers to "before AD 2000." The GICC05 chronology is based on counting annual layers, which are distinguishable due to seasonal variations of the ice (Andersen et al., 2006), and the sampling intervals along the core range from 1 a to 7 a.

Dating uncertainties were reported for each measurement of the raw data and they accumulate toward the more remote past, as a consequence of the layer-counting procedure, which starts from the top of the core (Svensson et al., 2008; Rasmussen et al., 2014). The dating uncertainties are reported to be roughly 5%, and reach a maximum counting error of 2573 a at an age of 59 420 a b2k. Note that there are no uncertainties in the relative timing of the $\delta^{18}$O and dust, since they are obtained from the same ice core.

The $\delta^{18}$O ratios are interpreted as proxies for surface air temperature variability, with algebraically higher ratios corresponding to warmer temperatures (Johnsen et al., 1992; Dansgaard et al., 1993; Johnsen et al., 2001a; Andersen et al., 2004). The dust concentrations have been proposed as proxies of large-scale atmospheric circulation changes, with higher particle counts being associated with stronger winds and thus with larger equator-to-pole temperature differences (Fischer et al., 2007; Steffensen et al., 2008). Although the two proxies thus represent very distinct climatic features, they exhibit a high degree of covariability

for the time interval 59 ka–22 ka b2k, as previously reported, e.g., by Johnsen et al. (1997); Ruth et al. (2003); Rasmussen et al. (2014). This covariability is best seen when considering the negative natural logarithm $-\log(\text{dust})$ of the data along with the $\delta^{18}$O data (Fig. 1C).

  The raw time series (blue curves in Figs. 1A,B) are preprocessed as follows: First, both time series are interpolated to an equidistant time axis with 5-a intervals. Second, gaps of varying length present in the dust time series and totaling about 6%

of the data points are filled by next-neighbor interpolation. Third, the multi-millennial trend is removed from both time series using a 40-ka running mean. Fourth, the noise level of the $\delta^{18}$O record is reduced by applying singular spectrum analysis (SSA), using the direct estimation of the covariance matrix as a Toeplitz matrix (Vautard et al., 1992; Ghil et al., 2002)). For the SSA smoothing, we use a window length of $M = 500$ a and retain the 16 leading reconstructed components. The SSA-smoothed time series (red curve in Fig. 1A) captures 60% of the total variance of the interpolated and detrended $\delta^{18}$O time

series.



## 2.2 Approximating the GLE in practice

It has recently been shown in general terms how to approximate the GLE (2) by a set of SDEs that is relatively easy to derive from the observables $\mathbf{x}$ (Kondrashov et al., 2015). Their MSMs both generalize EMR models and provide the correct time-continuous limit of such models. Based on their work, we directly derive here an approximation of (2) in terms of Eq. (3).

Compared to the GLE (2), the non-Markovian term with general kernel $\mathcal{G}$ is discretized and replaced by a sequence of Dirac kernels to obtain the second term on the right-hand side of Eq. (3), while the cubic term $D(\mathbf{x}, \mathbf{x}, \mathbf{x})$ guarantees the stability of the solutions. Essentially, the proposed model is thus a dynamical system with a twodimensional double-well potential that accounts for two alternative stable states. The additional memory term modulates the transitions between the two wells, which are stochastically forced by Gaussian white noise.

In practical applications, the SDE (3) is approximated by the following system of $k$ coupled, discrete difference equations with delays:

$$\mathbf{x}_{n+1} - \mathbf{x}_n = \left\{ A + \sum_{s=0}^{d} B_s \mathbf{x}_{n-s\tau} + C(\mathbf{x}_n, \mathbf{x}_n) + D(\mathbf{x}_n, \mathbf{x}_n, \mathbf{x}_n) \right\} \delta t_n + Q \left( \delta t_n \right)^{1/2} \chi_n. \tag{4}$$

Here $\mathbf{x}_n \in \mathbb{R}^k$ denotes the $k$-component observed variable $\mathbf{x}(t)$ at time $t = t_n$, with $1 \le n \le N$, and the $\chi_n \in \mathbb{R}^k$ denote $k$-dimensional, independent white-noise increments. Note that for modeling the NGRIP records, we have $k = 2$ variables given

by the $\delta^{18}O$ and dust measurements.

     In contrast to all other model parameters, the values for $d$ and $\tau$ used in the linear memory part in Eq. (4) are not varied in the parameter optimization: they are chosen in an outer loop so as to have the averaged PDFs of the model-simulated time series as close as possible to those of the observed ones. We choose $d = 2$ memory terms and a step width of $\tau = 75$ a, which leads to a total of 31 parameters to be estimated. Note that the total set of model parameters includes the standard deviations

of the noise residuals, as well as their correlation.

     The explicit coupled SDE system governing our stochastic-dynamic mode is hence given by





$$x_{n+1} - x_n = \left( A_1 + \sum_{s=0}^{2} \left( B_{11}^s x_{n-s\tau} + B_{12}^s y_{n-s\tau} \right) \right. \tag{5a}$$
$$+ C_{11} x_n^2 + C_{12} x_n y_n + C_{13} y_n^2$$
$$\left. + D_{11} x_n^3 + D_{12} x_n^2 y_n + D_{13} x_n y_n^2 + D_{14} y_n^3 \right) \delta t_n$$
$$+ \left( Q_{11} \xi_n^1 + Q_{12} \xi_n^2 \right) \delta t_n^{\,1/2},$$

$$y_{n+1} - y_n = \left( A_2 + \sum_{k=0}^{2} \left( B_{21}^k x_{n-k\tau} + B_{22}^k y_{n-k\tau} \right) \right. \tag{5b}$$
$$+ C_{21} x_n^2 + C_{22} x_n y_n + C_{23} y_n^2$$
$$\left. + D_{21} x_n^3 + D_{22} x_n^2 y_n + D_{23} x_n y_n^2 + D_{24} y_n^3 \right) \delta t_n$$
$$+ \left( Q_{21} \xi_n^1 + Q_{22} \xi_n^2 \right) \delta t_n^{\,1/2},$$

where $\mathbf{x_n} = (x_n, y_n)$ and $\tau = 75$ a.

## 2.3 Maximum Likelihood Estimation

Recently, Chorin and Lu (2015) introduced a general methodological framework for discrete stochastic parametrizations. In principle, an optimal parameter set $\Pi^*$ for the forms $A$, $B$, $C$, and $D$ of order $0 - 3$ can be determined by regressing the right-hand side of Eq. (4) onto the observed increments $\delta \mathbf{x}_n = \mathbf{x}_{n+1} - \mathbf{x}_n$, e.g. by ordinary least-squares (OLS) optimization:

$$\Pi^* = \arg\min_{\Pi} \left\| \frac{\delta \mathbf{x}_n}{\delta t_n} - \mathbf{F}^{\Pi}(\mathbf{x}_n, \dots, \mathbf{x}_{n-s\tau}) \right\|_2, \tag{6}$$

where $\mathbf{F}^{\Pi}$ denotes the operator corresponding to the right-hand side of (4), dropping the noise term and using a parameter combination $\Pi$.

We rely here, following, e.g., Kwasniok (2013); Chorin and Lu (2015); Krumscheid et al. (2015); Mitsui and Crucifix (2016), on Bayesian parameter inference for reduced stochastic models. For the present modeling task, we propose the Gaussian likelihood function

$$\mathcal{L}_{\mathcal{D}}(\Pi) = \prod_n \frac{1}{(2\pi)^2 |\Sigma|} \, \exp\left\{ -\frac{1}{2} \left( \frac{\delta \mathbf{x}_n}{\delta t_n} - \mathbf{F}^{\Pi}(\mathbf{x}_n, \dots, \mathbf{x}_{n-s\tau}) \right)^T \Sigma^{-1} \times \right.$$
$$\left. \left( \frac{\delta \mathbf{x}_n}{\delta t_n} - \mathbf{F}^{\Pi}(\mathbf{x}_n, \dots, \mathbf{x}_{n-s\tau}) \right) \right\}; \tag{7}$$

here $\Sigma$ is the covariance matrix of the noise, estimated from the residuals of the least-squares optimization.

Note that the functional form of the likelihood function in Eq. (7) assumes that the residuals are normally and independently distributed. The MZ formalism only assures that there exists a GLE with noise forcing that is uncorrelated with $\mathbf{x}$. The additional assumption that the noise is white in time is not theoretically guaranteed. Therefore, one has to check empirically how



well Gaussian white noise approximates the residual. Given that the stochastic difference equation (Eq. 4) only approximates the theoretical GLE provided by the MZ formalism, one also needs to validate empirically that the residuals are uncorrelated with the observations $\mathbf{x}$ (Kondrashov et al., 2015).

We remark that the approximation of the divided differences $\delta\mathbf{x}/\delta t$ by the function $\mathbf{F}^{\Pi}$ is in fact a multivariate multiple linear regression, with regressors chosen to be the polynomials in (5). It can easily be seen that MLE and OLS are equivalent for the case of univariate data and Gaussian errors, since the same term $\delta\mathbf{x}_n/\delta t_n - \mathbf{F}^{\Pi}$ is minimized in (6) and (7). In fact, subject to the assumption of uncorrelated — but not necessarily Gaussian — errors with zero mean and identical variance, the Gauss-Markov Theorem assures that parameters obtained from OLS, i.e. from Eq. (6), are the best linear unbiased estimators for such a regression. These estimators are *best* in the sense that they have lowest variance.

On the other hand, the MLE is asymptotically optimal. In particular, it is efficient in the sense that the variance of the parameter estimates achieves the so-called Cramér-Rao lower bound, which is optimal, as the number of samples tends to infinity (Andersen, 1970). Srivastava (1965) extended the Gauss-Markov Theorem to the multivariate case, where correlations among the variables lead to correlations between the error terms of the distinct variables, and thus result in cross-terms in the exponent of Eq. (7). Although both optimization approaches are thus equivalent in the case at hand, the MLE approach has the advantage that the parameter estimates can be interpreted as the most likely ones, given the observed data.

## 3   Results

As a training set for the parameter optimization of our stochastic-dynamic model in Eq. (5), we choose the time interval 59 ka– 22 ka b2k; this interval roughly coincides with Marine Isotope Stage 3 (approximately 60 ka–28 ka b2k). Our choice results in $N = 7528$ data points for each time series. The reason for this choice is that the layer-counted chronology has only been carried out until 59 ka b2k, and that the co-variability between $\delta^{18}$O and $\log(\mathrm{dust})$ is substantially reduced for the more recent part of the record, as already noticed by Ruth et al. (2002, 2003) and apparent in Fig. 1C here.

We use the natural logarithm of the dust time series (i) because of the large range of dust concentration values, and (ii) because it has high covariability with $\delta^{18}$O, cf. Fig. 1C; it also guarantees that the simulated dust values are positive. This logarithmic scale requires, however, high accuracy in the modeling of the dust concentrations to resolve the multiplicity of abrupt variations that span several orders of magnitude, cf. Fig. 1B.

Our results indicate that the residuals of the least-squares optimization (6) are indeed uncorrelated with the observations $\mathbf{x}$ — the correlations are less than $10^{-8}$ — and they are approximately Gaussian distributed (Fig. A1), while their autocorrelations decay very fast (not shown). These tests empirically support our choice of a Gaussian likelihood function of the form (7) for the MLE. Furthermore, these results allows us to integrate the stochastic-dynamic model given in Eq. (5) with an Euler-Maruyama scheme.

The specific values of the coefficients of (5), as derived via MLE, are given in Table A1. Note, in particular, the crucial nonlinear and lagged cross-interaction terms. For the reasons explained above, the parameters that minimize the least-squares





problem (6) are identical to the most likely parameter values as determined via MLE up to numerical precision (relative error less than $10^{-5}$).

In order to simulate optimal sample time series for the $\delta^{18}\mathrm{O}$ and $\log(\mathrm{dust})$ variables, the parameter combination $\Pi^*$ that maximizes the likelihood function (Eq. (7)) is used. The model is then integrated by the Euler-Maruyama scheme with uniform

step size of $\delta t = 10^{-5}$. The stochastic forcing is given by two-dimensional Gaussian white-noise increments multiplied by the Cholesky matrix Q. The residuals of the $\delta^{18}\mathrm{O}$ and $\log(\mathrm{dust})$ are correlated with a Pearson's correlation coefficient $r = -0.13$, resulting in a non-diagonal covariance matrix $\Sigma = \mathrm{QQ^T}$.

Illustrative time series of $\delta^{18}\mathrm{O}$ and $\mathrm{dust}$, simulated using the most likely model parameter combinations, are shown in Fig. 2. In Figs. 3A and 3B, the dashed lines show averages and uncertainties of PDFs of 1000 time series simulated using the most

likely parameter combinations. Figs. 3C and 3D show the average spectral densities of the latter time series. In this case, error bars would not be visible and are therefore omitted.

The means and standard deviations of the observed time series are reproduced well by the simulations: For the preprocessed $\delta^{18}\mathrm{O}$ the mean and standard deviation equal $-41.79 \pm 1.69$ compared to $-41.80 \pm 1.67$ for the simulations; the corresponding preprocessed and simulated values for $\log(\mathrm{dust})$ are $11.98 \pm 0.97$ and $11.99 \pm 0.93$. For the simulations, these values are

computed as averages over 1000 simulated sample time series, obtained using the parameter combinations that maximize the likelihood function (7).

The simulated time series (Fig. 2) exhibit abrupt changes that resemble the so-called Dansgaard-Oeschger events, which mark the sharp transitions from colder stadials to warmer interstadials (Dansgaard et al., 1993; Johnsen et al., 2001b; Rasmussen et al., 2014) in the original time series (Fig. 1). Given the more gradual temperature changes from interstadials to

stadials, the entire red curve in panel (A) of Fig. 2 has a dominant sawtooth-shape pattern. Recall that time here, as in Fig. 1, runs from right to left, as it does in Figs. 3(a,c) of (Krumscheid et al., 2015), which also display a high qualitative resemblance between their model-simulated and observed $\delta^{18}\mathrm{O}$ time series.

Following Krumscheid et al. (2015), we define the average waiting time $\tau_{\mathrm{DO}}$ between Dansgaard-Oeschger events as the sum of the average residence times in stadials and interstadials. For this purpose, stadials and interstadials are determined as

intervals for which the time series are, respectively, below or above the mean of the series. Due to their comparably high noise level, the high-resolution time series employed here are further smoothed by SSA using a window size of 500 a and keeping only the five leading reconstructed components. For the observed $\delta^{18}\mathrm{O}$ and $\log(\mathrm{dust})$, we obtain $\tau_{\mathrm{DO}} = 1506$ a and $1744$ a, respectively, compared to $\tau_{DO} = 1532 \pm 268$ a and $1497 \pm 360$ a for the simulations, computed as averages over 1000 simulated time series. Note that this definition of waiting times may not be optimal in view of the high noise level of the time series,

leading to different values for $\delta^{18}\mathrm{O}$ and $\log(\mathrm{dust})$, respectively. This definition is nevertheless employed here for the purpose of facilitating comparison with the results of (Krumscheid et al., 2015).

For the dust concentrations (Fig. 2B), there is a striking similarity between the observed and simulated time series in terms of episodes with low dust concentrations of variable durations, as well as the presence of burst episodes of variable magnitudes. Over the training interval, the preprocessed time series are correlated at $r = -0.86$, which equals the average correlation

between the simulated time series.





**Figure 2. Simulated $\delta^{18}$O and dust time series.** (A) Simulated $\delta^{18}$O time series (red). Gaussian white noise approximation of the residual removed during the SSA-smoothing of the original time series is added back in to obtain the full time series (blue). (B) Simulated dust time series, obtained as the exponential of the simulated $\log(\text{dust})$. (C) Simulated $\delta^{18}$O time series (red, same as in A) together with $-\log(\text{dust})$ (blue). Note that the strong co-variability between the two variables is captured very well by the model: Pearson's correlation coefficient, averaged over 1000 simulated time series, is $r = 0.86$.

The PDFs of the simulated $\delta^{18}$O and $\log(\text{dust})$ (red solid lines), obtained as averages over 1000 time series, are quite similar in shape to those of the preprocessed time series (blue solid lines) of both observed variables; see Figs. 3A and 3B. In particular, the bimodality of the PDFs, which reflects the relative persistence of stadials and interstadials, is reproduced by our inverse model. Uncertainties of the PDFs are derived on the basis of ensembles of 1000 simulated time series, produced by sampling from the most likely parameter combination. Error bars of the PDFs in Figs. 3A and 3B reflect the $2\sigma$ range of these ensembles.





The spectra of both the $\delta^{18}$O and the $\log(\text{dust})$ are well reproduced by our model, down to about a time periodicity of 100 a (Figs. 3C and 3D). The subcentennial periodicity in $\delta^{18}$O that is not captured by the model may be due to the preprocessing of the irregularly sampled data, and it will be discussed in future work.

## 4 Discussion

We studied the $\delta^{18}$O and dust time series obtained from the high-resolution NGRIP record for the 37 000-year interval 59 ka–22 ka b2k. The results described above show that the statistical properties of these time series — such as their PDFs, spectra, and average waiting times — can be approximated quite well by the proposed stochastic-dynamic model of Eq. (3).

The main features of our inverse model are (i) the nonlinear terms in the Markovian part; (ii) the inclusion of non-Markovian memory terms; and (iii) the coupling terms between the two time series. Cubic terms have previously been used to model the $\delta^{18}$O time series of the NGRIP record (Ditlevsen et al., 2007; Kwasniok, 2013; Krumscheid et al., 2015). However, non-Markovian contributions in an SDE model have, to the best of our knowledge, not been considered so far in the paleoclimate literature. Furthermore, we are not aware of modeling efforts that take advantage of the covariability between the $\delta^{18}$O and $\log(\text{dust})$ variables. It is shown in the following that the coupling terms between the two variables are crucial in order to capture the statistical characteristics of the measured NGRIP time series presented above, while the non-Markovian contribution is also significant.

The nonlinear terms can be physically motivated by the fact that the observed time series oscillate between two quasi-equilibria, namely the stadials and interstadials. If only linear terms were used, the bimodality of the observed time series, and hence the existence of two quasi-stable states, could not be reproduced (see Figs. A2 and A3).

Excluding the memory terms from our model leads to simulated time series for which the oscillations are too time-reversible, with the sawtooth shape of the $\delta^{18}$O series being either absent altogether or appearing for both an increase and a decrease in time, as seen in such a simulation for the intervals 50 ka–22 ka b2k and 59 ka–50 ka b2k, respectively (Fig. A4A). Moreover, the purely Markovian form of the model approximates the PDFs of the observed time series less well (Fig. A5). In particular, the bimodality of the PDFs for both $\delta^{18}$O and $\log(\text{dust})$ is weaker in this case, indicating that the memory terms do contribute to an appropriate modeling of the persistence in the stadials and interstadials, as well as of the transitions between them. In addition, the average waiting times between Dansgaard-Oeschger–like transitions are noticeably shorter in this case, namely $\tau_{\text{DO}} = 1505$ a for the $\delta^{18}$O and 1321 a for $\log(\text{dust})$.

When removing all coupling terms between the two variables from the inverse model (4), sawtooth-shaped transitions are completely absent from the simulated time series. In particular, the variations between quiet and burst episodes observed in the dust series are not reproduced in this case (Fig. A6). Furthermore, the bimodality of the PDFs is missed when excluding the couplings (Fig. A7), and the average waiting times are much too short, namely $\tau_{\text{DO}} = 1304$ a for the $\delta^{18}$O record and 553 a for the $\log(\text{dust})$ record.

Folowing previous authors (Kwasniok, 2013; Krumscheid et al., 2015; Mitsui and Crucifix, 2016), we also compare the different model versions in terms of the Bayesian (BIC) and Akaike (AIC) Information Criteria; these are defined as BIC =



**Table 1.** Bayesian Information Criteria (BIC) and Akaike Information Criteria (AIC) for the different model versions. Note that the model parameters include the standard deviations and correlation that appear in the respective model's noise term.

| Nonlinear | Memory | Coupling | # parameters | BIC | AIC |
|:---:|:---:|:---:|:---:|:---:|:---:|
| x | x | x | 31 | 2798.30 | 2562.34 |
|   | x | x | 17 | 2973.50 | 2844.07 |
| x |   | x | 23 | 3344.29 | 3169.11 |
| x | x |   | 14 | 3601.89 | 3495.30 |
|   |   | x | 9 | 4351.23 | 4282.66 |

$p\log(n+1)+2\log(\mathcal{L}^*)$ and AIC $= 2p(n+1)/(n-p)+2\log(\mathcal{L}^*)$, respectively. Here, $p$ denotes the number of model parameters, $n$ the total number of data points, and $\mathcal{L}^*$ the maximum value of the likelihood function (7); since we have two time series of length 7 529, $n = 2 \cdot 7\,529 = 15\,058$. The lower the value of BIC or AIC for a given model, the higher the relative confidence in that model.

For the case at hand, both BIC and AIC consistently favor the full model, which includes nonlinear, memory, and coupling terms. This is followed by the linear coupled model with memory terms, the nonlinear coupled model without memory terms, the nonlinear model with memory but without coupling terms, and finally the linear model without memory terms (Table 1).

   Note that the AIC penalizes higher numbers $p$ of model parameters less strongly than the BIC. Although the AIC was found, under certain conditions, to be optimal in selecting model candidates (e.g., Burnham and Anderson, 2002; Yang, 2005), notable
counterexamples are known (e.g., Penland et al., 1991).

   We thus suggest to interpret the values in Table 1 with caution. For example, in the case at hand both BIC and AIC propose higher confidence in the linear model than in the model without memory terms. This would suggest that the memory terms are more important than the higher-order parameters that correspond to the double-well shape of the potential. However, the PDFs of the cubic model without memory terms are considerably closer to the observed PDFs than the PDFs obtained from
the linear model including memory terms. In particular, by construction, the latter model cannot reproduce the bimodality of the observed PDFs. We would thus still argue that the nonlinear contributions are more important than the memory terms. Nevertheless, the presented AIC and BIC values provide information-theoretic evidence that the inclusion of memory terms does substantially improve the model.

   We emphasize that the full model proposed herein has the highest number of parameters out of the different candidates, but
is still the one with the lowest BIC and AIC. Therefore, it can be argued that this number of parameters is not too high, and it is not likely that the full model overfits the observed data.

   Furthermore, it should be noted that the values of BIC and AIC can only be compared on the basis of the same underlying data. Since we use a higher-resolution version of the NGRIP data as compared to the previous authors (Kwasniok, 2013; Krumscheid et al., 2015; Mitsui and Crucifix, 2016), the values for BIC and AIC presented here cannot be compared to the
values reported in those studies.





The model results presented here appear only in the high-resolution version of the NGRIP ice core record, which was originally sampled every 5 cm, a depth sampling that yielded temporal step sizes between one and seven years. For example, interpolating the raw data to a uniform grid with $\Delta t = 10$ a, instead of 5 a, leads to substantially less accurate approximations of the observed statistical properties of both the $\delta^{18}$O and dust time series (not shown). On the other hand, interpolating the raw data with a uniform sampling step of 3 a is problematic because of the original temporal step sizes, and does not further improve the results (not shown). It would thus appear that the 5-a uniform grid size is nearly optimal, given the irregularities in the sampling and the uncertainties in both the dating and the values of the records.

## 5   Conclusions

We have shown that a coupled, two-dimensional stochastic-dynamic model with cubic drift term and linear delay terms is capable of reproducing the statistical properties of $\delta^{18}$O and dust time series derived from the high-resolution NGRIP record for the interval 59 ka–22 ka b2k that roughly corresponds to Marine Isotope Stage 3. These statistical properties are expressed in terms of the PDFs of the time series, their power spectra, and the waiting times between sharp transitions from stadials to interstadials.

Key ingredients for an accurate simulation of the observed time series are:

1. High-resolution time series have to be used as training data, indicating that the high-frequency variability present in the records plays a vital role for the overall evolution of the climate processes that generated the NGRIP ice core. Interpolation of the raw data, which is sampled at depth intervals of 5 cm in the core, to 5-a intervals in the preprocessed time series was found to be optimal in our inverse-model setup. This finding is qualitatively consistent with the assertion of (Rypdal, 2016) that an increase in decadal-scale variability may be a statistical precursor for the abrupt transitions from stadials to interstadials during Dansgaard-Oeschger events.

2. Cubic terms need to be included in the Markovian part of the model. This can be physically motivated by the presence of two quasi-equilibria in the observed time series — the stadials and interstadials — that could not be modeled without two such quasi-stable states in the underlying dynamical system. Cubic terms have already been considered in previous attempts to model the $\delta^{18}$O time series of the NGRIP record (Ditlevsen et al., 2007; Kwasniok, 2013; Krumscheid et al., 2015); these attempts, however, did not include the dust series, used lower resolution data, and did not consider memory effects.

3. Coupling terms between the $\delta^{18}$O and dust variables are necessary to reproduce the sawtooth shape of the oscillations, with abrupt transitions from stadials to interstadials during Dansgaard-Oeschger events, followed by relatively slow relaxations back to the stadials.

4. Non-Markovian terms that account for memory effects are helpful. Their inclusion improves, in particular, the simulation of the sawtooth-shaped transitions between the cold and warm phases and of the bimodality of the PDFs.





Our results demonstrate that the statistical characteristics of the roughly 40-ka–long, high-resolution NGRIP time series of $\delta^{18}$O and dust considered here can be reproduced by a nonlinear inverse model without taking into account exogenous forcing, whether astronomical, solar or volcanic. There is thus no reason to assume that the temporal evolution of the $\delta^{18}$O ratios and dust concentrations — and hence that of the climatic variabilities they represent, in particular the transitions between stadials

and interstadials — are externally forced.

A quantitative analysis of the behavior of the observed and simulated time series under time reversal will be the subject of further research. Our results suggest that the coupling terms between $\delta^{18}$O ratios and dust concentrations, accompanied by memory terms in the linear part of the model, are needed for an accurate reproduction of the sharp transitions between stadials and interstadials, along with the slower relaxation back to the stadials. Furthermore, the predictive power of the proposed

stochastic-dynamic model for the abrupt transitions from stadials to interstadials should be addressed in future work.

## 6    Data availability

The high-resolution NGRIP data used in this study will be made available online once the paper is published.

## Appendix A:  Key ideas in deriving the GLE

The approach to data-driven stochastic-dynamic modeling taken here is rooted in the MZ formalism of statistical mechanics

(Zwanzig, 1964; Mori, 1965; Chorin et al., 2002; Chorin and Hald, 2013), which proposes an integro-differential closed form for model inference from partial data. While the derivation of such an MZ model does not allow one to easily simulate its solutions, it is possible, under suitable hypotheses, to obtain a good approximation thereof by a finite number of coupled SDEs (Chekroun et al., 2011b; Kondrashov et al., 2015).

Assume that $\mathbf{z}$ is a high-dimensional state vector, whose temporal evolution is governed by the following system of ordinary

differential equations, which is not explicitly known,

$$\frac{\mathrm{d}\mathbf{z}}{\mathrm{d}t} = \mathbf{F}(\mathbf{z}), \qquad \mathbf{z} \in \mathbf{R}^n, \tag{A1}$$

where $\mathbf{R}$ denotes the set of real numbers. Assume furthermore that $\mathbf{z}$ can be decomposed into a sum of an observed vector $\mathbf{x}$ and an unobserved vector $\mathbf{y}$, i.e. $\mathbf{z} = \mathbf{x} + \mathbf{y}$.

By orthogonally projecting (A1) onto the subspace spanned by the observed variables $\mathbf{x}$, via $P\mathbf{z} = \mathbf{x}$, we obtain

$$\frac{\mathrm{d}\mathbf{x}}{\mathrm{d}t} = P\mathbf{F}(\mathbf{x} + \mathbf{y}), \tag{A2}$$





which depends on the unobserved variable $\mathbf{y}$. By introducing the averaging

$$\overline{P\mathbf{F}(\mathbf{x})} := \int_Y P\mathbf{F}(\mathbf{x}+\mathbf{y})\,\mathrm{d}\mu_{\mathbf{x}}(\mathbf{y}), \tag{A3}$$

where $\mu_{\mathbf{x}}$ denotes the probability distribution of $\mathbf{y}$ conditioned on $\mathbf{x}$, we obtain

$$P\mathbf{F}(\mathbf{x}+\mathbf{y}) = \underbrace{\overline{P\mathbf{F}(\mathbf{x})}}_{\text{averaged part}} + \underbrace{\left(P\mathbf{F}(\mathbf{x}+\mathbf{y}) - \overline{P\mathbf{F}(\mathbf{x})}\right)}_{\text{fluctuating part}}. \tag{A4}$$

5    The parameterization of the fluctuating part in (A4) is at the core of any stochastic parameterization method — whether linear (Penland and Sardeshmukh, 1995) or nonlinear (Majda et al., 2001) — as well as of the MZ formalism.

Ergodic-type arguments show that the averaged part can in principle be learned from a time series, assuming the existence of a "nice" invariant measure (Chekroun et al., 2011a; Kondrashov et al., 2015):

$$\underset{f \in \mathcal{E}}{\mathrm{argmin}}\left(\lim_{T \to \infty} \frac{1}{T}\int_0^T \left\|\frac{\mathrm{d}\mathbf{x}}{\mathrm{d}t} - f(\mathbf{x}(t;\mathbf{y}_0))\right\|^2 \mathrm{d}t\right) = \overline{P\mathbf{F}(\mathbf{x})}, \tag{A5}$$

10    which holds for almost all $\mathbf{y}_0$ with respect to the Lebesgue measure; see Lemma 4.1 in (Kondrashov et al., 2015) for further details.

Note that Eq. (3) can be approximated by a Markovian SDE by using the Galerkin approximation techniques of (Chekroun et al., 2016) (Appendix C and Remark 5.1). In that respect, Eq. (3) can be put within an SDE format consistent with, although different from, the MSMs discussed by (Kondrashov et al., 2015).

15  **A1**

*Author contributions.* M.B., D.D.R., and A.S. provided the data. N.B., M.D.C., M.G., D.K., and H.L. conceived the research. N.B. conducted the numerical analysis and prepared the manuscript. All authors discussed the results, drew conclusions, and edited the manuscript.

*Competing interests.* The authors declare that they have no competing financial interests.

*Disclaimer.* TEXT





**Table A1.** The coefficients of the explicit SDDE system, obtained from MLE.

|  | $\delta^{18}\mathrm{O}(x)$ | $\mathrm{dust}(y)$ |
|---|---|---|
|  | $A_1 = -8.34 \times 10^6$ | $A_2 = 3.41 \times 10^7$ |
| $x_n$ | $B_{11}^0 = -4.50 \times 10^5$ | $B_{21}^0 = 1.51 \times 10^6$ |
| $y_n$ | $B_{12}^0 = 7.08 \times 10^2$ | $B_{22}^0 = 1.81 \times 10^3$ |
| $x_{n-\tau}$ | $B_{11}^1 = 8.37 \times 10^2$ | $B_{21}^1 = 2.76 \times 10^3$ |
| $y_{n-\tau}$ | $B_{12}^1 = 5.34 \times 10^5$ | $B_{22}^1 = -3.53 \times 10^6$ |
| $x_{n-2\tau}$ | $B_{11}^2 = 4.20 \times 10^3$ | $B_{21}^2 = 1.47 \times 10^4$ |
| $y_{n-2\tau}$ | $B_{12}^2 = 1.95 \times 10^3$ | $B_{22}^2 = 5.37 \times 10^3$ |
| $x_n^2$ | $C_{11} = -1.16 \times 10^4$ | $C_{21} = 2.80 \times 10^4$ |
| $x_n y_n$ | $C_{12} = -5.54 \times 10^3$ | $C_{22} = -6.78 \times 10^4$ |
| $y_n^2$ | $C_{13} = -5.60 \times 10^4$ | $C_{23} = 1.75 \times 10^5$ |
| $x_n^3$ | $D_{11} = -1.34 \times 10^2$ | $D_{21} = 2.25 \times 10^2$ |
| $x_n^2 y_n$ | $D_{12} = -4.48 \times 10^2$ | $D_{22} = -7.60 \times 10^1$ |
| $x_n y_n^2$ | $D_{13} = -1.38 \times 10^3$ | $D_{23} = 2.63 \times 10^3$ |
| $y_n^3$ | $D_{14} = 6.61 \times 10^1$ | $D_{24} = -1.76 \times 10^3$ |
| $\xi_n^1$ | $Q_{11} = 69.47$ | $Q_{21} = -12.97$ |
| $\xi_n^2$ | $Q_{21} = 0$ | $Q_{22} = 99.57$ |

*Acknowledgements.* This research was initiated by a collaboration between D.D.R. and Sigfús Johnsen (RIP), to whose memory it is dedi-
cated. N.B. acknowledges funding by the Alexander von Humboldt Foundation and the German Federal Ministry for Education and Research.
M.D.C., M.G., D.K. and H.L. acknowledge support by grant N00014-16-1-2073 from the Multidisciplinary University Research Initiative
(MURI) of the Office of Naval Research and by National Science Foundation grant OCE-1243175. M.D.C. and H.L. also acknowledge
support by National Science Foundation grants DMS-1616981 and DMS-1616450, respectively. D.K. also acknowledges support by the
Government of the Russian Federation (agreement #14.Z50.31.0033 with the Institute of Applied Physics of RAS). This is LDEO xxxx
contribution. NGRIP is directed and organized by the Ice and Climate research group, Niels Bohr Institute, University of Copenhagen. It
is supported by funding agencies in Denmark (FNU), Belgium (FNRS-CFB), France (IPEV and INSU/CNRS), Germany (AWI), Iceland
(RannIs), Japan (MEXT), Sweden (SPRS), Switzerland (SNF) and the USA (NSF, Office of Polar Programs).



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





**Figure 3. Statistical properties of the observed and simulated $\delta^{18}$O and dust time series.** (**A**) PDFs for the raw, preprocessed, simulated $\delta^{18}$O. Note how the bimodality, corresponding to the transitions between stadials and interstadials, only arises after reducing the noise level in the time series by means of SSA, leading to the preprocessed time series. (**B**) PDFs for the raw, preprocessed, and simulated $\log(\text{dust})$ time series. PDFs are obtained as averages over 1000 simulated time series, each obtained from the most likely model parameters (solid red). Error bars indicate the $2\sigma$-range of uncertainties derived from theses 1000 sample time series. (**C**) Power spectral densities for the interpolated, smoothed, and simulated $\delta^{18}$O. (**D**) Power spectral densities for the preprocessed and simulated $\log(\text{dust})$. The spectral densities are smoothed using the multitaper method (Vautard et al., 1992). For the spectra, the corresponding uncertainties would be hardly visible and are therefore omitted.





Figure A1: Probability density of the least-squares residuals for $\delta^{18}$O (blue) and $\log(\mathrm{dust})$ (red). The residuals for the $\log(\mathrm{dust})$ can be approximated very well by a Gaussian, but also for the $\delta^{18}$O the approximation is sufficiently good to justify the choice of a Gaussian Likelihood function for the maximum likelihood estimation.





Figure A2: Same as Fig. 2 in the main text, but for the model without nonlinear terms.





Figure A3: Same as Figs. 3A and 3B in the main text, but for the model without nonlinear terms.





Figure A4: Same as Fig. 2 in the main text, but for the model without memory terms.





Figure A5: Same as Figs. 3A and 3B in the main text, but for the model without memory terms.





Figure A6: Same as Fig. 2 in the main text, but for the model without coupling terms. Note that a co-evolution of the observed $\delta^{18}O$ and $\log(\text{dust})$ time series can in this case not be expected to be reproduced.





Figure A7: Same as Figs. 3A and 3B in the main text, but for the model without coupling terms.