# Peer review of "Inverse stochastic-dynamic models for high-resolution Greenland ice-core records"

_Earth System Dynamics, 2017_

## Referee Comment (RC1) · Anonymous Referee #1 · 1 Feb 2017

This is a model study finding the best fit of a stochastic difference model with delay to the joined isotope and dust records from the NGRIP ice core. The model is a 2D Langevin equation with additional time delay terms. This is not a new approach, but it introduces in a systematic way the dating uncertainty into the model fitting procedure. Furthermore, the effect of memory (non-Markovian terms in the Langevin equation) for improving the fit is shown. The only part I am not completely convinced by is the statement that the coupling between d18O and dust explains the sawtooth shape of the time series (time asymmetry) (point 3 in conclusions). I have seen an earlier version of the manuscript, and this new version is improved and expanded to accommodate my previous comments. The science is sound and the paper is well written. I can recommend publication without further revisions.

---

## Referee Comment (RC2) · T. Mitsui (Referee) · 5 May 2017

**1   General comments**

This work is a contribution to inverse stochastic-dynamic modeling of abrupt climate changes, so-called Dansgaard-Oeschger (DO) events, using high-resolution Greenland ice core records. While many of the previous stochastic-dynamic models of DO events have been introduced in intuitive, ad-hoc ways (for example, assuming a double-well potential or an additive white noise), their stochastic delay differential equation is grounded in the Mori-Zwanzig (MZ) formalism of statistical physics. It is interesting to consider the coupling between different proxies $\delta^{18}O$ and log(dust) though its physical mechanism is not mentioned in this paper. The authors show that the coupling and the

memory terms due to MZ formalism are important to capture the asymmetric sawtooth oscillations and, in part, they are contributing to the bimodal character of the probability density functions (PDFs). These points are particularly valuable in inverse modeling of DO events. On the other hand, as mentioned below, some remarks are based on visual assessments, and thus sound less convincing. There are also confusions in the use of information criteria. Therefore, I recommend the publication of this article if the following points are revised.

**2  Specific comments**

(1) The successful reproduction of the time-reversal asymmetry of the sawtooth oscillations is a key result of this article. However, the quantitative assessment of the time-reversal asymmetry has not been done. The visual comparison between Fig. 2 and Fig. 4A has to be complemented by a more quantitative assessment. For instance, one may claim that simulated events during 56–58 ka and 29–30 ka in Fig. 2 look time-symmetric. I suggest some quantitative assessment of the time-reversal asymmetry (for example, by using the third-order moment $x(t)x^2(t+\tau) - x^2(t)x(t+\tau)$ as in Kwasniok, 2013).

Also, the authors write *"In particular, the variations between quiet and burst episodes observed in the dust series are not reproduced in this case (Fig. A6)"*, but I recognize quiet episodes during 34–38 ka, 42–45 ka, and 47–50 ka and burst episodes during the others in Fig. A6.

(2) There are confusions in the use of information criteria BIC and AICc[1]. If $n$ is the number of time points of observation vectors, they should be BIC$= p\log(n) - 2\log(L^*)$ and AICc$= 2pn/(n-p-1) - 2\log(L^*)$, where $p$ is the number of parameters and $L^*$
* * *
[1]I suggest to mention that the sample-size-corrected AIC, AICc, is used in this paper instead of the standard AIC.

is the maximum likelihood. The authors seem to follow the notation in Krumscheid et al., 2015, but in their paper, the number of observations is $n + 1$. Also the sign in front of $2\log(L^*)$ should be minus. The number $n$ in BIC is the number of time points of observation vectors, and not the total number of the elements in the observation vectors (e.g. Raftery 1995; see Yamaguchi and Higuchi 2006 for the case of state-space model). Thus, the authors should not multiply the number of time points 7529 by 2 to obtain the value of $n$ used for BIC.

(3) The number of memory terms $d = 2$ and the delay $\tau = 75$ a are chosen such as to have the average PDFs of the simulated time series as close as possible to those of the observed ones. However, given that the purely Markovian form of the model approximates the PDFs of the observed time series relatively well (Fig. A5), I'm not sure that matching PDFs is an appropriate way to choose the values of $d$ and $\tau$. Does BIC or AICc select similar values?

(4) A comment about the conclusion: *"Our results demonstrate that the statistical characteristics of the roughly 40-ka–long, high-resolution NGRIP time series of $\delta^{18}O$ and dust considered here can be reproduced by a nonlinear inverse model without taking into account exogenous forcing, whether astronomical, solar or volcanic. There is thus no reason to assume that the temporal evolution of the $\delta^{18}O$ ratios and dust concentrations – and hence that of the climatic variabilities they represent, in particular the transitions between stadials and interstadials – are externally forced."*

I agree that abrupt DO warmings and coolings themselves are not governed by external forcings, but, for example, Mitsui and Crucifix (2017) show the influence of external forcings on the DO events. In that paper, the slow decay of the sample autocorrelation function of the NGRIP $\log_{10}Ca^{2+}$ record is simulated only in the presence of ice volume forcing. The BIC evidence of forced models against unforced model is also reported for some model classes. Thus, there are some reasons to assume that the temporal

evolution of the $\delta^{18}$O ratios and dust concentrations have a non-autonomous character.

**3  Technical corrections and minor comments**

Page 1, Line 16: Is "consation" right?

Page 2, Line 10: $F$ should be bold as in Eq. (1)?

Page 6, Line 7: twodimensional → two-dimensional

Page 6, Line 12: If $n$ and $s$ are both non-dimensional indices, it is strange that $\tau$ (=75 a) is dimensional in Eqs. (4)–(7).

Page 7, Line 20: A square root $\sqrt{\cdot}$ is missing for $(2\pi)^2|\Sigma|$ in Eq. (7)?

Page 8, Lines 5 and 31: (5) → Eq. (5) is desirable?

Page 9, Line 5: what is the unit for $\delta t = 10^{-5}$?

Page 11, Line 21: Fig. A4"A" is referred to in the text, but "A" is not labeled on the corresponding panel.

Page 11, Line 10: Although the authors mention that *"non-Markovian contributions*

*in an SDE model have, to the best of our knowledge, not been considered so far in
the paleoclimate literature"*, it is considered by Pelletier (2003) for glacial-interglacial
cycles (cf. non-Markovian 'deterministic' models are considered by Rial (2004) for DO
oscillations and by Berger (1999) for glacial-interglacial cycles).

The number of observation vectors is 7529 according to Page 12, Line 3, but it is
$N = 7528$ according to Page 8, Line 19.

The caption of Figure 3C refers to "interpolated", "smoothed", and "simulated", but its
legend refers to "interpolated", "preprocessed", and "simulated".

Figures 3C and 3D: What are the number of tapers and the time-bandwidth parameter
used to estimate these spectra? The spectra seem strongly smoothed.

A reference (Mitsui and Crucifix, 2016) is updated (please see below).

Some of the previous models of DO events exhibit self-sustained oscillations. How
much noises are important to generate DO transitions in this model? Do oscillations
disappear if the noise covariance matrix $Q$ is set to zero?

**References**

Kwasniok, F.: Analysis and modelling of glacial climate transitions using simple dynamical sys-
   tems, Philosophical Transactions of the Royal Society of London A: Mathematical, Physical
   and Engineering Sciences, **371**, 20110472, 2013.
Krumscheid, S., Pradas, M., Pavliotis, G.A., and Kalliadasis, S.: Data-driven coarse graining in
   action: Modelling and prediction of complex systems, Physical Review E, **92**, 042139, 2015.

Raftery, A.E.: Bayesian Model Selection in Social Research, Sociological Methodology, **25**, 111–163, 1995.

Yamaguchi, R. and Higuchi, T.: State-space approach with the maximum likelihood principle to identify the system generating time-course gene expression data of yeast, Int. J. Data Mining and Bioinformatics, **1**, 77–87, 2006.

Pelletier, J.D.: Coherence resonance and ice ages, Journal of Geophysical Research, **108**, 4645, 2003.

Rial, J.A.: Abrupt climate change: chaos and order at orbital and millennial scales, Global and Planetary Change, **41**, 95–109, 2004.

Berger, W.H., The 100-kyr ice-age cycle: Internal oscillation or inclinational forcing?, Journal Earth Sciences, **88**, 305–316, 1999.

Mitsui, T. and Crucifix, M.: Influence of external forcings on abrupt millennial-scale climate changes: a statistical modelling study, Climate Dynamics, **48**, 2729–2749, 2017.
* * *

---

## Author Comment (AC1) · 22 May 2017

**Response to Anonymous Referee #1:**

We thank the reviewer for his positive evaluation of our manuscript. Regarding the relevance of coupling terms between the $\delta^{18}$O and dust variables, we acknowledge that their effect on the temporal asymmetry in terms of sawtooth-shaped oscillations between stadials and interstadials is hard to assess quantitatively. Coupling terms are, however, strongly supported by the model selection criteria (AICc and BIC) despite their introducing a substantial number of additional parameters (14 vs. 31). Furthermore, including coupling terms is necessary for an accurate approximation of the statistical properties of the NGRIP time series. In the revised manuscript, we will replace item 3 of

the conclusions by "Including coupling terms between the $\delta^{18}$O and dust variables sub-stantially improves the statistical characteristics of the simulated time series in terms of reproducing the bimodality of the probability density functions as well as the correct average waiting time between subsequent transitions from stadials to interstadials."

---

## Author Comment (AC2) · 22 May 2017

**Responses to Takahito Mitsui:**

We thank you for this thorough evaluation of our manuscript. Your detailed comments and suggestions have been very helpful in our revision.

Regarding your Specific Comments:

1. Following your suggestion, we have computed the third-order moment,

$$M(\theta) = \langle x(t)x^2(t+\theta) - x^2(t)x(t+\theta)\rangle_t \,,$$

as done by Kwasniok et al. (2013) to quantify the time-reversal asymmetry of simulated time series. Fig. 1B shows that, for delays $\theta$ up to roughly 1000 years, $M(\theta)$ computed from dust simulations does exhibit a somewhat similar behavior as the $M(\theta)$ computed from the observed dust. Quantitatively, the similarity between $M(\theta)$ computed for the observed and simulated dust, respectively, is also supported by Kendall's $\tau$ (red curve in Fig. 2D). The temporal asymmetry is, however, not reproduced by $\delta^{18}O$ simulations (Fig. 1A), as also confirmed by the blue curve in Fig. 2D.

Given the very strong correlations $r^P$ between the two variables, with $r^P \approx 0.9$ for both observations and simulations, the discrepancies between the resulting $M(\theta)$ given $\delta^{18}O$ or dust data are rather surprising. These discrepancies suggest that $M(\theta)$ might not be the best measure for quantifying whether sawtooth-shaped oscillations are present or not.

Still, in terms of the $M$-score, the reproduction of the time-reversal asymmetry is not as successful as we inferred visually from comparing Figs. 2 and 4A in the original manuscript. We do acknowledge that the behavior of single simulated time series can be misleading, and will modify our conclusions accordingly, to include a paragraph on the quantitative results obtained on the basis of the third-order moment $M(\theta)$.

The main contribution of including memory terms into the model is to improve the average simulated waiting times between subsequent transitions from stadials to interstadials, cf. Fig. 2E, for $55$ a $\lesssim \tau \lesssim 60$ a. See also our response in item 3. below. The memory terms also help improve the probability density functions (PDFs) of simulated time series, as shown by a comparison of Figs. 3A and 3B to Fig. A5 in our original manuscript. The improvement is particularly noteworthy in reproducing the bimodality of the PDF of the $\delta^{18}O$ time series. Furthermore, the AICc and BIC criteria support, in general, the models that do include memory terms.
2. You are right, there were several mistakes in the stated formulae for both AICc and BIC, and we do appreciate your pointing them out. Apart from typos, we indeed took $n$ to be the total number of data points, although it should be the number of (two-dimensional) observation vectors. Correcting for this mistake leads to slightly smaller values for AICc and BIC than stated in our manuscript, but the relative order of AICc and BIC values, corresponding to the different model candidates, remains unchanged. Therefore, the conclusions drawn from both model selection criteria are unaltered.

3. From a theoretical perspective, based on the Mori-Zwanzig formalism, there is no need to further increase the number of memory terms $d$ because, for $d = 2$, the residuals are already white in space and time. This point is mentioned in the manuscript, but we will emphasize it further in the revision.

   The appropriate length of the delay $\tau$ could, of course, be directly determined based on BIC or AICc criteria. For any $\tau$ such that $0 < \tau \leq 1000$ time steps (of $5$ years), both selection criteria consistently favor the non-Markovian model, with the memory terms, over the Markovian one, without them. The AICc and BIC criteria, however, exhibit lowest values for $\tau = 1$ time step (5 years), and monotonically increase for longer $\tau$, whilst staying below the respective values obtained for $\tau = 0$; see Fig. 2A.

   Choosing $\tau = 5$ years (i.e. one time step) would, however, lead to less accurate approximations of the statistical properties of the observed NGRIP time series than either no memory or longer memory; see Figs. 2B–2E. The reproduction of the statistical characteristics of the NGRIP time series is thus not optimal for the value of $\tau$ that is suggested by AICc and BIC, but for considerably longer memory. Values of $\tau$ between $55$ and $80$ years (i.e., between $11$ and $16$ time steps) yield comparably good approximations when all statistical characteristics are taken into account.

   In particular, a rather narrow range of possible memory step sizes, namely

$\tau = 55$ a or $\tau = 60$ a, yields an accurate approximation of the average waiting times between subsequent transitions from stadials to interstadials. In the revised version, we will include these observations to explain and motivate the right choice of the memory parameter, for which a value of $\tau = 60$ a seems to be optimal when taking into account the AICc, together with the statistical characteristics shown in Figs. 2B–2E.

4. We acknowledge that including ice-volume forcing improves the slow decay of the sample autocorrelation function of the NGRIP $\log(\text{Ca}^{2+})$ record in your recent paper, and that the BIC supports external forcing for some of the studied models. We also agree that the relevance of external forcing cannot be ruled out by our results. In the conclusion, we merely wanted to make the point that a relatively simple model can approximate the dynamics of the NGRIP time series quite well without external forcing, as long as memory and couplings between oxygen isotope and dust are considered. We believe that the extent to which external forcing might contribute to the longer-term evolution of DO cycles remains open and subject to further inquiry. We will rephrase the corresponding sentences in the conclusions accordingly.

Regarding your Technical corrections and minor comments: In the revised version, we will address all of them following your suggestions:

- P1, L16: This should have been "consideration".

- P2, L10: We agree!

- P6, L7: Correct, thank you!

- P6, L12: The memory-window length $\tau$ is 15 time steps and thus indeed non-dimensional. In the text, we translated this to $75$ a because each time step corresponds to $5$ a. We will clarify this in the revised manuscript.

- P7, L20: Agreed!

- P9, L5 and L31: Yes, we agree.

- P9, L5: The step size for the numerical integration of our models (using the Euler-Maruyama scheme) is $\delta t = 10^{-5}$, which is thus non-dimensional. Compared to the observational data, this step size would correspond to $5$ a. Essentially, we thus rescaled time in order to guarantee a stable numerical integration. In the revised version of our manuscript, this will be pointed out more explicitly.

- P11, L21: We will add the labels in the revised manuscript.

- P11, L10: Thank you for pointing us to these interesting references. We were not aware of them and will cite them in the revised manuscript.

- The total number of data points is $N = 7529$, but this leaves $7528$ difference quotients to be fitted. We'll add a note on this in the revision.

- Yes, we will correct this in the revised manuscript.

- For the multitaper estimate of the PSD, we set the time-halfbandwidth parameter (NW) to a value of 4, corresponding to $2 \cdot \text{NW} - 1 = 7$ tapers. The PSD shown for the simulations is an average over 500 simulated time series, and is therefore strongly smoothed.

- We will update the reference to your paper in Climate Dynamics in the revised version.

- The drift term in our model corresponds to two stable states (stadial and interstadial conditions), and the parameters are kept fixed. Transitions between the two stable states are thus solely triggered by fluctuations. By setting the entries of the noise covariance matrix $Q$ to zero, the system would thus stay at either one of the fixed points without any further dynamics.

We will revise our manuscript in accordance with these responses once the editor agrees.

Figure captions:

**Fig. 1.** Third-order statistical moment $M(\theta) = \langle x(t)x^2(t+\theta) - x^2(t)x(t+\theta) \rangle_t$ for the observed NGRIP time series (solid blue), the full model including memory terms with step size $\tau = 60$ a (solid red), and the model without memory terms (dashed red). Note that for increasing delays $\theta$ the values of $M(\theta)$ are affected more and more by the non-stationarity of the data, and should therefore be interpreted with care. The upper panel (A) shows $M(\theta)$ for the $\delta^{18}$O time series, while the lower one (B) shows $M(\theta)$ for the dust time series.

**Fig. 2.** A. Log-likelihood and AICc for different values of the memory step size $\tau$. The AICc is computed as AICc $= 2pn/(n-p-1) - 2\log\mathcal{L}^*$. B. Difference between observed and simulated standard deviations. C. $L^2-$ and $L^\infty-$distances between observed and simulated probability density functions. D. Kendall's $\tau$ between the third-order moments $M(\theta)$, computed for observed and simulated time series, respectively. E. Difference between observations and simulations in terms of average waiting times between subsequent transitions from stadials to interstadials. In B–E, statistics for simulated time series are obtained as averages over $400$ simulations using the full model.

[Figure]

A

B

$M(\theta)$

$M(\theta)$

time lag $\theta$ $[a]$

$\delta^{18}$O preprocessed
$\delta^{18}$O simulated (full)
$\delta^{18}$O simulated (no memory)

$-\log(\text{dust})$ preprocessed
$-\log(\text{dust})$ simulated (full)
$-\log(\text{dust})$ simulated (no memory)

**Fig. 1.** see text for caption

[Figure]

**Fig. 2.** see text for caption

---

## Author Response (AR1)

**Point-by-point responses to the referees' comments**

**Response to Anonymous Referee # 1:**

We thank the reviewer for his positive evaluation of our manuscript. Regarding the relevance of coupling terms between the $\delta^{18}O$ and dust variables, we acknowledge that their effect on the temporal asymmetry in terms of sawtooth-shaped oscillations between stadials and interstadials is hard to assess quantitatively. Coupling terms are, however, strongly supported by the model selection criteria (AICc and BIC) despite their introducing a substantial number of additional parameters (14 vs. 31). Furthermore, including coupling terms is necessary for an accurate approximation of the statistical properties of the NGRIP time series. In the revised manuscript, we have replaced item 3 of the conclusions by "Coupling terms between the $\delta^{18}O$ and dust variables substantially improve the statistical characteristics of the simulated time series: The reproduction of the bimodality of the PDFs, as well as of the correct average waiting time between subsequent transitions from stadials to interstadials, are substantially improved when coupling terms are included."

**Responses to Takahito Mitsui:**

We thank you for this thorough evaluation of our manuscript. Your detailed comments and suggestions have been very helpful in our revision.

We would like to point out here that in addition to changes that we made following your suggestions, we have also changed the method to reduce the noise level of the $\delta^{18}O$ record: instead of smoothing the time series using SSA, we use a simple Butterworth low-pass filter in the revised version. The reason for this is that we have realized that the SSA-based smoothing leads to possibly spurious peaks in the high-frequency part of the PSD (compare Fig. 3C in the first version of our manuscript to Fig. 3C in the revised version), which should be avoided. Quantitatively, the reported values for standard deviations, correlations, BIC, AICc, etc., have changed slightly due to this modification, but the qualitative conclusions remain unaltered.

Regarding your Specific Comments:

1. Following your suggestion, we have computed the third-order moment,
$$M(\theta) = \langle x(t)x^2(t + \theta) - x^2(t)x(t + \theta)\rangle_t \,,$$

as done by Kwasniok et al. (2013), to quantify the time-reversal asymmetry of simulated time series. Fig. 4B of the revised manuscript shows that, for delays $\theta$ up to 1000 years, $M(\theta)$ computed from dust simulations does exhibit a somewhat similar behavior as the $M(\theta)$ computed from the observed dust. Quantitatively, the similarity between $M(\theta)$ computed for the observed and simulated dust, respectively, is also supported by Kendall's $\tau$ (red curve in Fig. 5D of the revised manuscript). The temporal asymmetry is, however, not reproduced by $\delta^{18}O$ simulations (Fig. 4A of the revised manuscript), as also confirmed by the blue curve in Fig. 5D of the revised manuscript.

Given the very strong correlations $r^P$ between the two variables, with $r^P \approx 0.85$ for both observations and simulations, the discrepancies between the resulting $M(\theta)$ given $\delta^{18}O$ or dust data are rather surprising. These discrepancies suggest that $M(\theta)$ might not be the best measure for quantifying whether sawtooth-shaped oscillations are present or not.

Still, in terms of the $M$-score, the reproduction of the time-reversal asymmetry is not as successful as we inferred visually from comparing Figs. 2 and 4A in the first version of the manuscript. We do acknowledge that the behavior of single simulated time series can be misleading, and have modified our discussion and conclusion accordingly, to include a paragraph on the quantitative results obtained on the basis of the third-order moment $M(\theta)$.

The main contribution of including memory terms into the model is to improve the average simulated waiting times between subsequent transitions from stadials to interstadials, cf. Fig. 5E of the revised manuscript, for $55$ a $\lesssim \tau \lesssim 60$ a. See also our response in item 3. below. The memory terms also help improve the probability density functions (PDFs) of simulated time series, as shown by a comparison of Figs. 3A and 3B to Fig. A5. The improvement is particularly noteworthy in reproducing the bimodality of the PDF of the $\delta^{18}O$ time series. Furthermore, the AICc and BIC criteria support, in general, the models that do include memory terms.

2. You are right, there were several mistakes in the stated formulae for both AICc and BIC, and we do appreciate your pointing them out. Apart from typos, we indeed took $n$ to be the total number of data points, although it should be the number of (two-dimensional) observation vectors. Correcting for this mistake leads to different values for

AICc and BIC than stated in our manuscript, but the relative order of AICc and BIC values, corresponding to the different model candidates, remains unchanged. Therefore, the conclusions drawn from both model selection criteria are unaltered. Note, in particular, that the modified smoothing of the $\delta^{18}O$ time series (Butterworth low-pass filter instead of SSA) leads to different values of AICc and BIC as compared to the first version of our manuscript.

3. From a theoretical perspective, based on the Mori-Zwanzig formalism, there is no need to further increase the number of memory terms $d$ because, for $d = 2$, the residuals are already uncorrelated with the observations $\mathbf{x}$. This point is mentioned in the manuscript, but we have emphasized it further in the revision.

   The appropriate length of the delay $\tau$ could, of course, be directly determined based on the AICc or BIC criteria. For any $\tau$ such that $0 < \tau \leq 1000$ time steps (of 5 years), both selection criteria consistently favor the non-Markovian model, with the memory terms, over the Markovian one, without them. The AICc and BIC criteria, however, exhibit lowest values for $\tau = 1$ time step (5 years), and monotonically increase for longer $\tau$, whilst staying below the respective values obtained for $\tau = 0$; see Fig. 5A of the revised manuscript.

   Choosing $\tau = 5$ years (i.e. one time step) would, however, lead to less accurate approximations of the statistical properties of the observed NGRIP time series than either no memory or longer memory; see Figs. 5B–5E. The reproduction of the statistical characteristics of the NGRIP time series is thus not optimal for the value of $\tau$ that is suggested by AICc and BIC, but for considerably longer memory. Values of $\tau$ between 55 and 80 years (i.e., between 11 and 16 time steps) yield comparably good approximations when all statistical characteristics are taken into account.

   In particular, a rather narrow range of possible memory step sizes, namely $\tau = 55$ a or $\tau = 60$ a, yields an accurate approximation of the average waiting times between subsequent transitions from stadials to interstadials. In the revised version, we will include these observations to explain and motivate the right choice of the memory parameter, for which a value of $\tau = 60$ a seems to be optimal when taking into account the AICc, together with the statistical characteristics shown in Figs. 5B–5E.

4. We acknowledge that including ice-volume forcing improves the slow

decay of the sample autocorrelation function of the NGRIP $\log(\mathrm{Ca}^{2+})$ record in your recent paper, and that the BIC supports external forcing for some of the studied models. We also agree that the relevance of external forcing cannot be ruled out by our results. In the conclusion, we merely wanted to make the point that a relatively simple model can approximate the millennial-scale dynamics of the NGRIP time series quite well without external forcing, as long as memory and couplings between oxygen isotope and dust are considered. We believe that the extent to which external forcing might contribute to the longer-term evolution of DO cycles remains open and subject to further inquiry. We have rephrased the corresponding sentences in the conclusions accordingly.

Regarding your Technical corrections and minor comments: In the revised version, we will address all of them following your suggestions:

- P1, L16: This should have been "consideration".

- P2, L10: We agree!

- P6, L7: Correct, thank you!

- P6, L12: The memory-window length $\tau$ is 15 time steps and thus indeed non-dimensional. In the text, we translated this to 75 a because each time step corresponds to 5 a. We have clarified this in the revised manuscript, where we now argue that a memory step size of 60 a is optimal.

- P7, L20: Agreed!

- P9, L5 and L31: Yes, we agree.

- P9, L5: The step size for the numerical integration of our models (using the Euler-Maruyama scheme) is $\delta t = 10^{-5}$, which is thus non-dimensional. Compared to the observational data, this step size would correspond to 5 a. Essentially, we thus rescaled time in order to guarantee a stable numerical integration. In the revised version of our manuscript, this is pointed out more explicitly.

- P11, L21: The reference to the specific labels is removed in the revised version of the manuscript.

- P11, L10: Thank you for pointing us to these interesting references. We were not aware of them and will cite them in the revised manuscript.

- The total number of data points is $N = 7529$, but this leaves 7528 difference quotients to be fitted. We have added a note on this in the revision.

- Yes, we have corrected this in the revised manuscript.

- For the multitaper estimate of the PSD, we set the time-halfbandwidth parameter (NW) to a value of 4, corresponding to $2 \cdot \text{NW} - 1 = 7$ tapers. The PSD shown for the simulations is an average over 1000 simulated time series, and is therefore strongly smoothed.

- We have updated the reference to your paper in Climate Dynamics in the revised version.

- The drift term in our model corresponds to two stable states (stadial and interstadial conditions), and the parameters are kept fixed. Transitions between the two stable states are thus solely triggered by fluctuations. By setting the entries of the noise covariance matrix $Q$ to zero, the system would thus stay at either one of the fixed points without any further dynamics.

[revised manuscript text omitted]

---

## Author Response (AR2)

**Responses to the second review by Takahito Mitsui:**

We thank you for your very thorough evaluation of our revised manuscript. We have followed all your suggestions, as can be seen in the track-changes document below.

[revised manuscript text omitted]